# Layer-specific proteomic analysis of human hearts in patients with sudden cardiac death

**Yu Kakimoto** [1]*, **Xueting Guan**[1], **Atsushi Ueda**[1], **Yayoi Kimura**[2], **Tomoko Akiyama**[2], **Masayuki Tanaka** [3], **Haruka Ikeda**[1], **Kazuho Maeda**[4], **Shotaro Isozaki**[1]

**1** Department of Forensic Medicine, Tokai University School of Medicine, Isehara, Kanagawa, Japan, **2** Advanced Medical Research Center, Yokohama City University, Kanazawa-ku, Kanagawa, Japan, **3** Department of Life Science Support, Research Innovation Center, University Hospitals Sector, Tokai University, Isehara, Kanagawa, Japan, **4** Department of Legal Medicine, Yokohama City University Graduate School of Medicine, Kanazawa-ku, Kanagawa, Japan

* kakimoto@tokai.ac.jp

## Abstract

### Background

Recent studies have shown that decreased longitudinal strain in speckle-tracking echography can predict sudden cardiac death (SCD) in patients with cardiac hypertrophy (CH). Histologically, the heart wall consists of the inner longitudinal, middle circular, and outer longitudinal layers. Thus, layer-specific proteomic changes may contribute to SCD risk.

### Methods

The three layers in human cardiac tissues were obtained during autopsies of SCD, compensated CH, and control cases (18 cases aged > 40 years, 54 samples in total). SCD cases consisted of patients with ischemic or hypertensive heart failure, whereas CH and control cases were accidental deaths. After histological analysis, cardiomyocytes were collected separately from the three layers of the left ventricular wall using laser microdissection. The extracted proteins were analyzed using liquid chromatography–tandem mass spectrometry, followed by label-free quantification.

### Results and discussion

Histologically, cardiomyocytes were enlarged in all layers of SCD and CH cases without significant progression of myocardial fibrosis. The proteomic profiles of SCD and CH cases were distinguishable from those of control cases, especially in the inner layer. The levels of mitochondrial and calcium-handling proteins were significantly decreased in SCD hearts. Arrhythmogenic changes, including decreased PKP2 and RYR2 levels, developed in a stepwise manner from control through CH to SCD, most prominently in the inner layer. Immunohistochemical analysis showed reduced levels

**Data availability statement:** All relevant data for this study are publicly available from the Dryad repository (https://doi.org/10.5061/dryad.q573n5tx2).

**Funding:** This work was supported by JSPS KAKEHNHI (grant number: 23H03178) and the MEXT promotion of the Distinctive Joint Research Center Program at the Advanced Medical Research Center, Yokohama City University (grant number: JPMXP0622717006).

**Competing interests:** The authors have declared that no competing interests exist.

**Abbreviations:** BNP, brain natriuretic peptide; CH, cardiac hypertrophy; KO, conditional knockout; FFPE, formalin-fixed paraffin-embedded; GO, gene ontology; ID, intercalated disk; KO, knockout; MS, mass spectrometry; ROS, reactive oxygen species; SCD, sudden cardiac death; SR, sarcoplasmic reticulum.

of PKP2 in intercalated disks and RYR2 in sarcomeres. Because proteomic alterations precede the progression of myocardial fibrosis, their detection may enhance the accuracy of postmortem diagnosis of SCD in middle-aged and older asymptomatic individuals with CH.

## 1. Introduction

Sudden cardiac death (SCD) accounts for millions of deaths worldwide every year [1,2] and exerts substantial psychological and economic effects on society [3,4]. To date, many clinical trials have been performed for the risk stratification of SCD in patients with cardiomyopathies, those with inherited arrhythmias, and the general population [5,6]. However, SCD prevention remains challenging because SCD may occur in apparently healthy individuals, and the pathologies underlying SCD are heterogeneous.

Recently, imaging techniques have been further developed, and speckle-tracking echocardiography has advanced the quantification of regional left ventricular function by discerning the multidirectional components of ventricular deformation [7]. High-resolution imaging can detect mortality risk from an early phase as reduced longitudinal strain predicts SCD in patients with hypertrophic cardiomyopathy [8], as well as cardiovascular mortality in hypertensive patients [9] and even the general population [10,11]. Regional differences in the three-dimensional contractility of the heart can reflect the spatial heterogeneity of myocardial pathologies. However, the anisotropic proteomic changes of the human heart remain unknown.

Anatomically, the ventricular wall comprises a three-dimensional mesh of longitudinal and circular cardiomyocytes [12]. The structure of the myocardial layers—inner longitudinal layer (In), middle circular layer (Mid), and outer longitudinal layer (Out)—is most distinct at the mid-level of the left ventricle. The coordinated function of these anisotropic layers is crucial for maintaining stable systemic circulation, and layer-specific proteomic alterations can underlie the physiological adaptation and pathological degeneration of the heart. We recently performed layer-specific proteomic profiling of normal human hearts, and demonstrated the anisotropic levels of actin-filament-binding proteins. CTNNA3, highly expressed in the inner layer, can stabilize the endocardial conduction system, and MYH1, highly expressed in the outer layer, can support cardiac contractile endurance [13].

In the current study, we focused on layer-specific proteomic alterations in cases of SCD and cardiac hypertrophy (CH). In middle-aged and older adults, CH is often associated with obesity and hypertension and is an established risk factor for SCD [14,15]. Sudden death cases are often examined at forensic autopsy, including SCD cases with lethal CH, and accidental death cases with CH not contributing to the cause of death, termed compensated CH. However, the postmortem differentiation between lethal and compensated CH is sometimes difficult due to the lack of specific histological findings, and arrhythmia-induced deaths, which account for more than 50% of SCD cases, are often determined by exclusion [16]. Thus, molecular markers

for lethal arrhythmogenesis are not only necessary to determine the probability of SCD in patients but also to accurately diagnose the cause of death postmortem.

In this study, we hypothesized that layer-specific proteomic alterations occur in SCDs of middle-aged and older adults with CH. We performed high-throughput mass spectrometry (MS) on cardiomyocytes collected separately from three myocardial layers. Spatial proteomic analysis revealed various proteomic changes that occurred mainly in the inner layer and proceeded from CH to SCD, preceding the progression of myocardial fibrosis.

## Materials and methods

### Study population

Forensic cases in which the deceased were discovered within 24 h after their death were selected for this study. All patients were over 40 years of age and were divided into three groups: SCD, CH, and control (Fig 1A). SCD was defined as death within 1 h of symptom onset or unwitnessed death with the individual exhibiting normal health within 24 h before death [1]. SCD cases in this study died from either ischemic or hypertensive heart failure, whereas CH and control cases were accidental deaths. Causes of death were comprehensively determined based on clinical history, macroscopic autopsy findings, microscopic pathological examination, blood biochemistry, and toxicology screening. Ischemic heart failure was diagnosed based on the presence of coronary atherosclerosis with > 80% luminal stenosis. Hypertensive heart failure was diagnosed based on a clinical history of hypertension and cardiomegaly, defined as a heart weight/body height ratio of > 2.4 g/cm [15]. Accidental death was defined as death due to fatal trauma in the absence of significant disease or intoxication. CH cases were considered as compensated hypertrophy in the absence of a clinical history of heart failure or cardiac pathology other than cardiomegaly. Patients with hypertrophic cardiomyopathy were excluded. Control individuals were enrolled in a previous proteomic study [13]; all samples including those from controls were simultaneously analyzed in this study. Written informed consent for the academic use of the tissue was obtained from the relatives of all decedents included in the study, and the study was approved by the institutional ethics committee (approval number: 21R177). The recruitment period was from May 2, 2014, to December 31, 2024. This study was conducted in accordance with the ethical standards of the 1964 Declaration of Helsinki and its subsequent amendments.

### Histological analysis

The body of the deceased was stored at 4°C until cardiac tissue sampling during the autopsy. The heart was transversely sectioned at the mid-level between the base and apex, and preserved in 10% formalin for approximately 48 h. Formalin-fixed paraffin-embedded (FFPE) tissues were used for subsequent analyses. The three myocardial layers were demarcated on transverse sections of the left ventricular free wall under microscopy. Specifically, cross-sectioned myocardial bundles were identified as the inner and outer longitudinal layers, and longitudinally-sectioned myocardial bundles were considered the middle circular layer. Diagonally-sliced or partly disarrayed myocardial bundles were excluded from analysis. Microscopic measurements were performed on 4-μm sections of the left ventricular free wall. The minor diameter of the cardiomyocytes at the nuclear point was measured at 200× magnification using hematoxylin and eosin staining, and the values were averaged from 10 viewing fields per case. As we assessed 10 cardiomyocytes per field, a total of 100 cardiomyocytes were analyzed per myocardial layer in each case. Myocardial fibrosis was assessed at 100× magnification using picrosirius red staining, excluding the endocardium, epicardium, and chordae tendineae. The fibrosis rate was quantified using ImageJ software (https://imagej.net/), and the percentage of the total image was averaged over 20 viewing fields per case [17]. Multiple groups were compared using the Steel–Dwass test in Excel Statistics 2015 (SSRI, Tokyo, Japan), and a $p$-value < 0.05 was considered significant.

### Laser microdissection

The 5-μm FFPE sections of the left ventricular free wall were mounted on PET-membrane frame slides (Micro Dissect GmbH, Herborn, Germany). After staining with toluidine blue, the slides were air-dried. Cardiomyocytes covering an area

**A**

|  | Sudden cardiac death | Cardiac hypertrophy | Control |
|---|---|---|---|
| **Case** | 8 males | 5 meles | 5 males |
| **Age** (y.o.) | 55 $\pm$ 10 | 58 $\pm$ 6 | 54 $\pm$ 4 |
| **Body weight** (kg) | 81.4 $\pm$ 12.8 | 79.6 $\pm$ 6.5 | 70.3 $\pm$ 9.2 |
| **BMI** (kg/m2) | 26.8 $\pm$ 4.1 | 27.3 $\pm$ 0.8 | 23.6 $\pm$ 2.4 |
| **Comorbidities** (based on available information) | Hypertension (4) Diabetes (1) Sleep apnea syndrome (2) | Hypertension (2) Diabetes (1) | — |
| **BNP** (pg/mL) | 13.6 $\pm$ 18.0 | 7.3 $\pm$ 5.4 | 3.4 $\pm$ 1.8 |
| **Heart weight** (g) | 499 $\pm$ 92** | 493 $\pm$ 51* | 349 $\pm$ 24 |
| **FFPE stored time** (months) | 63.9 $\pm$ 37.1 | 46.4 $\pm$ 30.8 | 50.4 $\pm$ 16.7 |

**B**

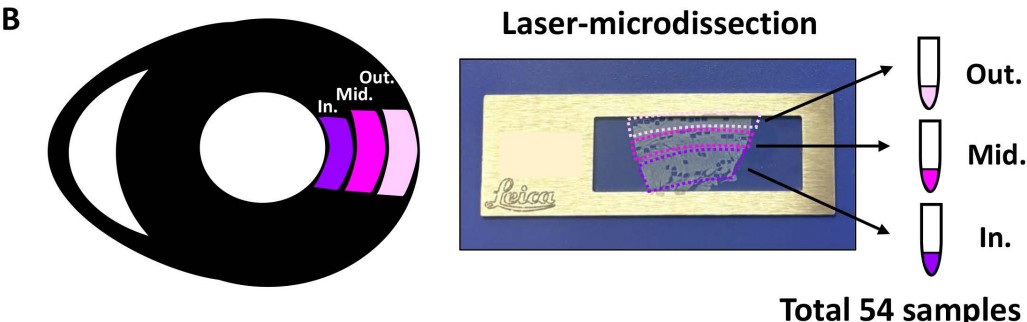

**Fig 1. Microscopic tissue sampling of three myocardial layers in the human heart. (A) Case characteristics.** Data are presented as the mean ± standard deviation. * $p < 0.05$, ** $p < 0.01$ compared with control as determined via the Steel-Dwass test. (B) Laser microdissection of the three myocardial layers. Cardiomyocytes were separately collected from each layer. FFPE, formalin-fixed paraffin-embedded; FFPE stored time, interval from paraffin embedding to protein extraction.

of 10 mm² were dissected from each myocardial layer of the specimens using laser microdissection (LMD 6500; Leica Microsystems, Wetzlar, Germany; Fig 1B). Tissues with large capillaries, fibrosis, or lipid accumulation were excluded.

## Sample preparation for proteomic analysis

Microdissected samples were washed with 150 μL of 100 mM ammonium bicarbonate containing 30% acetonitrile and then dissolved in 50 μL of the same solution. The samples were then heated at 95°C for 90 min, cooled on ice, and centrifuged at 5,000 × $g$ for 1 min. Subsequently, samples were enzymatically digested with 5 μL of 0.2 μg/μL Trypsin Gold Mass Spectrometry Grade (Promega, Madison, WI, USA) at 37°C for 16 h. After centrifugation at 15,000 × $g$ for 3 min, the peptides in the collected supernatants were quantified using a DC Protein Assay Kit II (Bio-Rad, Hercules, CA, USA). In total, 5 μg of peptide samples were dried using a SpeedVac Vacuum Concentrator (Thermo Fisher Scientific, Waltham,

MA, USA) and dissolved in 100 μL of 2% acetonitrile containing 0.1% trifluoroacetic acid. Samples were manually cleaned up using GL-Tip SDB (GL Science Inc., Tokyo, Japan), according to the manufacturer's instructions, but the solution volume was changed from 20 μL to 100 μL. Samples were eluted with 100 μL of 30% and 60% acetonitrile containing 0.1% trifluoroacetic acid, respectively. The eluted peptides were dried using a Speed Vac concentrator.

## Proteomic analysis using data dependent acquisition-MS analysis

Proteome profiles were individually determined by proteomic analysis using liquid chromatography–tandem MS and label-free quantitation. The desalted peptides were resuspended in 2:98 [v/v] acetonitrile/water and 0.1% formic acid. Then, 0.5 μg of each sample was injected individually and analyzed once using a Q Exactive mass spectrometer (Thermo Fisher Scientific) equipped with an UltiMate 3000 LC system (Thermo Fisher Scientific). The mass spectrometer was operated using Xcalibur software. Peptides were loaded on a trap column (100 μm × 20 mm, C18, 5 μm, 100 A, Thermo Fisher Scientific) and subsequently separated on a Nano HPLC capillary column (75 μm × 180 mm, C18, 3 μm; Nikkyo Technos Co., Ltd., Tokyo, Japan) with a linear gradient of 2–33% buffer B (95:5 [v/v] acetonitrile/water, 0.1% formic acid) in buffer A (2:98 [v/v] acetonitrile/water, 0.1% formic acid) for 120 min at a flow rate of 300 nL/min. Full-scan MS spectra were measured from 300 to 1,500 $m/z$, and each of the 20 most abundant ions was subsequently subjected to HCD product ion scans. Tandem MS scanning conditions were as follows: spray voltage, 2.0 kV; capillary temperature, 250°C; and normalized collision energy, 27%.

Label-free quantitation was conducted using Progenesis QI for proteomics (version 4.2; Nonlinear Dynamics). The samples were categorized into three groups—In, Mid, and Out—and subsequently statistically analyzed. For protein and peptide identification, peak lists were generated using Progenesis QI for proteomics and then searched against human protein sequences in the UniProtKB/Swiss-Prot database (Jan 2020) using Mascot software (v2.7, Matrix Science, London, UK). The search parameters were set as follows: trypsin digestion with two permitted missed cleavages; peptide mass tolerance of ± 5 ppm; fragment mass tolerance of ± 0.05 Da; and common variable modifications including methionine oxidation, cysteine carbamidomethylation, protein N-terminal acetylation, and N-terminal carbamylation. A 1% overall false discovery rate was applied as the threshold to export the results from the database search using the Mascot software. Additionally, peptides with a peptide ion score ≥ 30 were utilized for quantitation. Only peptides uniquely assigned to a single protein were included in quantitative analysis. Those shared with other protein hits were excluded. Significance was determined using a $p$-value < 0.05 in analyses of variance of the normalized values among groups.

## Bioinformatic analysis

Principal component analysis was performed using SIMCA software (Infocom, Tokyo, Japan). Gene Ontology (GO) enrichment analysis was performed using R (4.4.2) and R Studio (2024.09.0 + 375). Prior to GO enrichment analysis, the accession IDs of significantly different proteins were extracted. Using AnnotationDbi (1.68.0), these IDs were converted to Entriz IDs and gene symbols from the org.Hs.e.g.,db database (3.20.0). Converted probes were subjected to GO enrichment analyses with a focus on the "Molecular Function" category using the package *clusterProfiler* (4.14.0) [18]. The Benjamini–Hochberg method was applied to adjust the $p$-value, with a significance cutoff of 0.05. The threshold of the q-value was set to 0.05 to further minimize the false discovery rate. Kyoto Encyclopedia of Genes and Genomes (KEGG) pathway analysis was performed using the same parameters. Dot and box plots were created using the package *ggplot2* in the framework *tidyverse* (2.0.0) [19]. To illustrate the connections between genes and pathways, a network plot was created using the packages *igraph* (2.1.1) and *ggraph* (2.2.1). The Kruskal–Wallis test was used to assess statistical significance in the three groups, followed by pairwise comparisons using the Steel–Dwass test (R v.4.4.2). The Friedman test was used for inter-layer comparisons within each group, followed by pairwise comparisons using the Wilcoxon signed-rank test. $p < 0.05$ was deemed to indicate statistical significance.

## PKP2 sequencing

DNA was extracted from cardiac tissue samples using a QIA Amp DNA Investigator Kit (Qiagen, Hilden, Germany). Oligo-nucleotide primers (S1 Table) were designed for the amplification and sequencing of *PKP2* exons. PCR amplification was performed using Tks Gflex DNA Polymerase (Takara Bio Inc., Kusatsu, Japan) with 10 ng of DNA as the template. The PCR products were purified using ExoSAP-IT Express (Thermo Fisher Scientific). The directly sequenced products were analyzed using a Genetic Analyzer 3500 (Thermo Fisher Scientific). For all SCD cases, the coding region of *PKP2* was determined, and no mutations affecting *PKP2* level were found.

## Immunohistochemical analysis

The 4-μm sections of the left ventricular free wall were used for immunohistochemistry. Because intercalated disks (IDs), where PKP2 localizes, and T-tubules, where RYR2 localizes, are clearly visible in longitudinal sections of cardiomyocytes, we used vertical sections in addition to transverse sections of the ventricular wall. A vertical section block was available for four SCD cases, three CH cases, and three control cases. Antigen retrieval was conducted by heating the samples in Target Retrieval Solution (pH 9.0; Dako, Santa Clara, CA, USA), at 95°C for 20 min. Specimens were immunostained with an anti-PKP2 antibody (1:800; ab223757, Abcam, Cambridge, UK) for 60 min at room temperature and an anti-RYR2 antibody (1:200, 19765–1-AP, Proteintech, Rosemont, IL, USA). Subsequently, samples were incubated with a secondary antibody (Histofine MAX-PO-MULTI; Nichirei Biosciences, Tokyo, Japan) at room temperature for 30 min. DAB and hematoxylin were used for color development.

# Results

## Myocardial hypertrophy without significant fibrosis in SCD and CH cases

Body mass index and serum brain natriuretic peptide (BNP) levels were not clinically different among all cases, whereas heart weight and cardiomyocyte diameter were significantly increased in the SCD and CH groups (Fig 1A and 2A). The heart weight/body height ratio was > 2.4 g/cm in all SCD and CH cases. Slight myocardial fibrosis was observed in the inner and middle layers of cases with SCD and CH; however, myocardial fibrosis was not significantly different among the three study groups (Fig 2B). No significant inflammatory infiltrate was observed in any samples.

## Pronounced proteomic alterations in the inner layer of SCD cases

In total, 54 samples (three cardiac layers of 18 enrolled cases) were collected and analyzed. The proteomic profiles of SCD and CH cases were distinguishable from those of the control cases in all layers; however, the characteristic changes of SCD cases were most prominent in the inner layer (Fig 3A, S2 Table). In all layers of SCD cases, the numbers of proteins with decreased levels were larger than those of proteins with increased levels. Of the 1,250 proteins identified in the inner layer, 26 were significantly increased in SCD cases, and 111 were significantly decreased compared with control cases. Of the 1,217 proteins identified in the middle layer, 13 were significantly increased and 90 were significantly decreased. Of the 1,255 proteins identified in the outer layer, 13 were significantly increased and 87 were significantly decreased. Proteins with the greatest changes in the three groups are shown in Fig 3B.

In contrast, the differences between SCD and CH cases were minimal. Only 17 (10 increased and 7 decreased), 12 (9 increased and 3 decreased), and 13 (5 increased and 8 decreased) proteins were differentially detected in the inner, middle, and outer layers, respectively (S2 Table). The inter-layer comparisons in each group were separately presented in S3-S5 Table, and the list of differentially expressed proteins were summarized in S6 Table. Note that the original data for inter-layer comparisons in the control cases was previously published [13]; in the present study, we reanalyzed these data using multiple nonparametric methods.

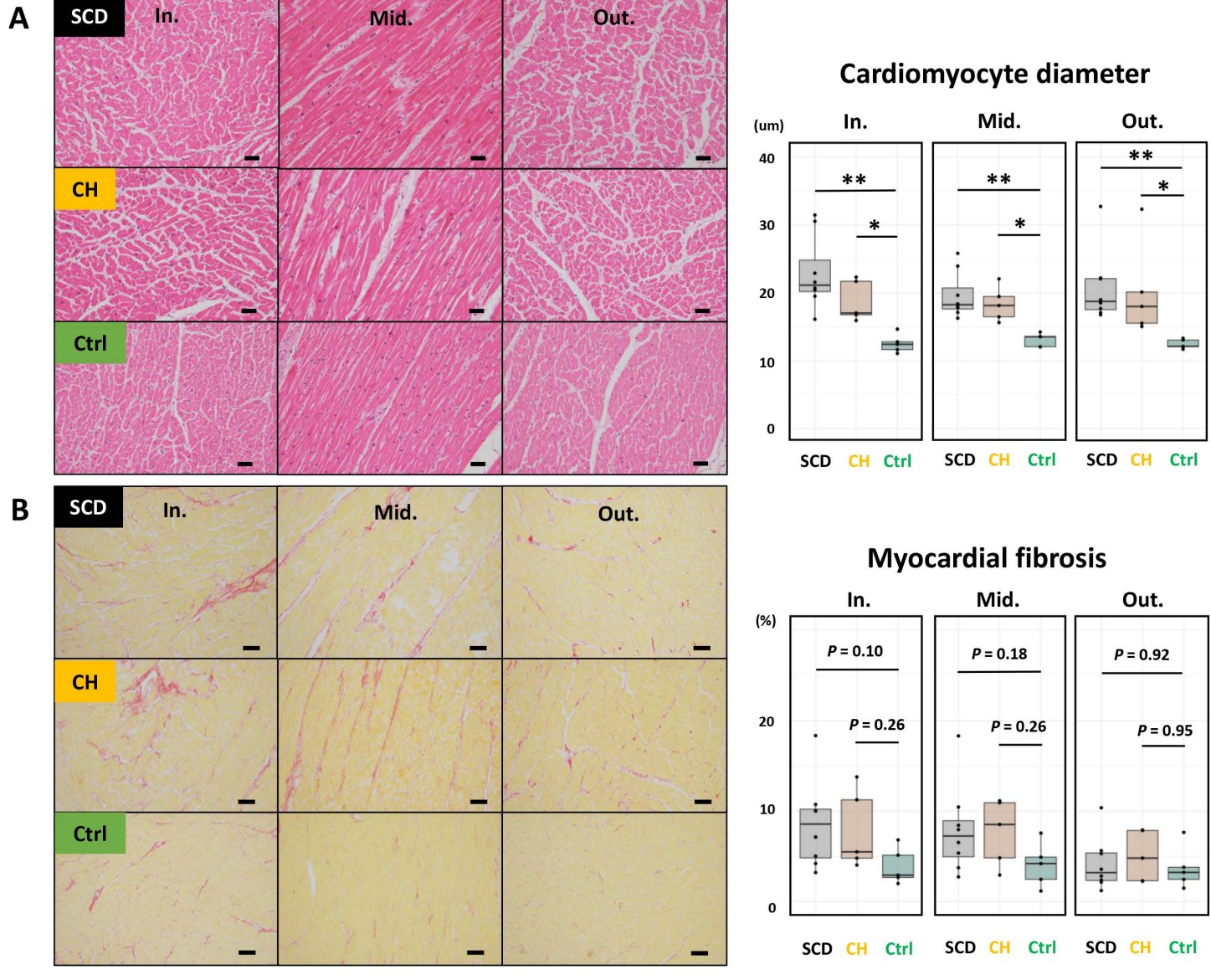

**Fig 2. Histological findings in human cardiac tissue samples.** Cardiomyocyte diameters are increased in SCD and CH cases compared with controls, whereas myocardial fibrosis does not differ significantly among the groups. Representative images of the three layers stained with hematoxylin and eosin (A; scale bar: 50 μm) or picrosirius red (B; scale bar: 100 μm) dye. *$p < 0.05$, **$p < 0.01$ (Steel–Dwass test). Abbreviations: CH, cardiac hypertrophy; Ctrl, control; SCD, sudden cardiac death.

Many GO terms related to molecular function were associated with decreased proteins in SCD cases compared with control cases (S7 Table). In contrast, no GO terms were associated with increased proteins in either the inner or middle layers in SCD cases, and only five terms were identified for increased proteins in the outer layer. The major GO terms related to molecular function are summarized in Fig 4A. Cadherin-binding proteins, transmembrane transporter proteins, and mitochondrial proteins with oxidoreductase or electron transfer activities were significantly decreased in all layers of SCD cases, most prominently in the inner layer. Nucleosomal proteins were significantly increased in the outer layer of SCD cases.

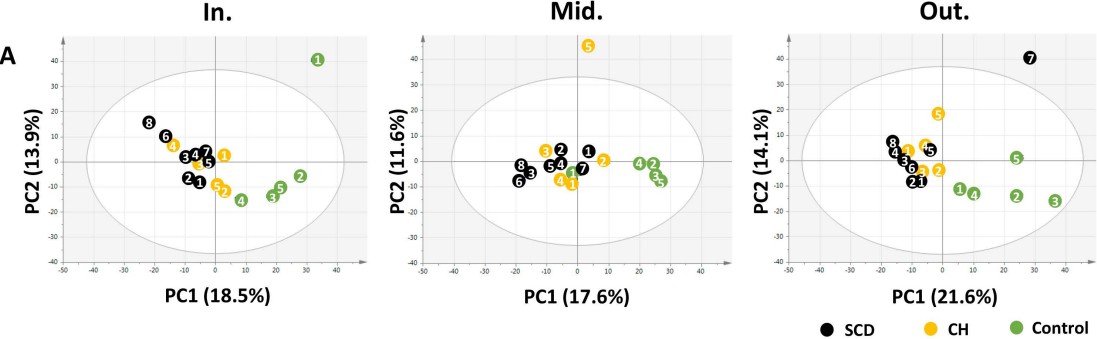

**A**

**In.** — PC2 (13.9%) vs PC1 (18.5%)
**Mid.** — PC2 (11.6%) vs PC1 (17.6%)
**Out.** — PC2 (14.1%) vs PC1 (21.6%)

● SCD ● CH ● Control

**B**

| Fold Change (SCD vs Ctrl) | Gene ID | Description | Relative Abundance | | |
|---|---|---|---|---|---|
| | | | SCD | CH | Ctrl |
| **In.↓** | | | | | |
| 9.20 | YWHAB | 14-3-3 protein beta/alpha | 152,022 | 231,007 | 1,398,769 |
| 3.10 | MLIP | Muscular LMNA-interacting protein | 502,199 | 958,322 | 1,561,732 |
| 2.19 | CRIP2 | Cysteine-rich protein 2 | 1,103,134 | 1,800,960 | 2,420,146 |
| 2.02 | HSDL2 | Hydroxysteroid dehydrogenase-like protein 2 | 5,869,900 | 8,664,733 | 11,838,139 |
| 2.00 | CDS2 | Phosphatidate cytidylyltransferase 2 | 923,613 | 1,573,170 | 1,844,200 |
| 1.80 | H2AZ1 | Histone H2A.Z | 1,315,407 | 1,372,133 | 2,364,069 |
| 1.72 | CSTB | Cystatin-B | 2,761,862 | 3,539,610 | 4,739,892 |
| 1.70 | NDUFA7 | NADH dehydrogenase [ubiquinone] 1 alpha subcomplex subunit 7 | 2,506,025 | 3,581,283 | 4,262,329 |
| 1.70 | NACA | Nascent polypeptide-associated complex subunit alpha, muscle-specific form | 3,342,104 | 3,686,679 | 5,681,546 |
| 1.66 | MICOS13 | MICOS complex subunit MIC13 | 485,907 | 619,511 | 805,839 |
| **Mid.↓** | | | | | |
| 5.16 | YWHAB | 14-3-3 protein beta/alpha | 200,976 | 364,035 | 1,230,079 |
| 3.22 | CRIP2 | Cysteine-rich protein 2 | 765,710 | 1,446,331 | 2,663,023 |
| 2.68 | MLIP | Muscular LMNA-interacting protein | 491,794 | 836,976 | 1,458,554 |
| 2.50 | SYNPO | Synaptopodin | 419,201 | 503,988 | 1,157,461 |
| 2.49 | CLIC4 | Chloride intracellular channel protein 4 | 237,528 | 381,573 | 639,774 |
| 2.38 | SERPINC1 | Antithrombin-III | 320,571 | 474,091 | 665,079 |
| 2.38 | MT-ND4 | NADH-ubiquinone oxidoreductase chain 4 | 1,442,133 | 2,868,665 | 3,484,032 |
| 2.13 | PANK4 | 4'-phosphopantetheine phosphatase | 152,949 | 232,004 | 355,582 |
| 1.87 | VAPA | Vesicle-associated membrane protein-associated protein A | 1,615,893 | 2,249,213 | 3,249,617 |
| 1.81 | GPD1L | Glycerol-3-phosphate dehydrogenase 1-like protein | 1,889,183 | 2,226,549 | 3,519,840 |
| **Out.↓** | | | | | |
| 6.81 | WFS1 | Wolframin | 27,510 | 61,726 | 187,387 |
| 2.45 | SNX3 | Sorting nexin-3 | 206,072 | 235,641 | 504,877 |
| 2.36 | CDS2 | Phosphatidate cytidylyltransferase 2 | 838,615 | 1,371,250 | 1,977,471 |
| 2.14 | MLIP | Muscular LMNA-interacting protein | 1,140,961 | 1,533,769 | 2,437,359 |
| 2.09 | MT-ND4 | NADH-ubiquinone oxidoreductase chain 4 | 1,581,595 | 2,264,900 | 3,310,852 |
| 2.02 | HHATL | Protein-cysteine N-palmitoyltransferase HHAT-like protein | 1,152,569 | 1,265,528 | 2,330,466 |
| 1.93 | FAM162A | Protein FAM162A | 2,123,878 | 2,911,905 | 4,101,117 |
| 1.89 | CSTB | Cystatin-B | 1,853,378 | 2,663,515 | 3,494,504 |
| 1.79 | H2AZ1 | Histone H2A.Z | 1,328,001 | 1,414,730 | 2,373,471 |
| 1.69 | DCXR | L-xylulose reductase | 2,798,825 | 3,097,709 | 4,730,465 |
| **In.↑** | | | | | |
| 17.91 | ABHD14B | Protein ABHD14B | 28,070 | 20,726 | 1,568 |
| 4.26 | RPL17 | 60S ribosomal protein L17 | 166,458 | 137,821 | 39,118 |
| 2.89 | DAG1 | Dystroglycan | 520,338 | 415,912 | 180,889 |
| 1.79 | AMBP | Protein AMBP | 3,059,085 | 2,246,408 | 1,708,587 |
| 1.74 | SMYD1 | Histone-lysine N-methyltransferase SMYD1 | 3,481,364 | 3,022,891 | 1,999,293 |
| 1.65 | C4A | Complement C4-A | 3,844,255 | 2,728,739 | 2,328,043 |
| 1.44 | RPS12 | 40S ribosomal protein S12 | 749,583 | 660,212 | 518,987 |
| 1.23 | GBE1 | 1,4-alpha-glucan-branching enzyme | 9,842,045 | 8,779,202 | 8,020,599 |
| 1.16 | RPS2 | 40S ribosomal protein S2 | 1,185,693 | 1,126,801 | 1,026,971 |
| **Mid.↑** | | | | | |
| 2.33 | PSMA6 | Proteasome subunit alpha type-6 | 506,281 | 380,058 | 232,527 |
| 2.07 | PGM5 | Phosphoglucomutase-like protein 5 | 15,829,951 | 11,537,211 | 8,029,759 |
| 1.37 | RAB2A | Ras-related protein Rab-2A | 1,687,326 | 1,679,673 | 1,191,173 |
| **Out.↑** | | | | | |
| 2.29 | MTX2 | Metaxin-2 | 908,143 | 679,459 | 396,412 |
| 2.11 | CAPNS1 | Calpain small subunit 1 | 884,104 | 670,186 | 418,616 |
| 2.03 | IDH1 | Isocitrate dehydrogenase [NADP] cytoplasmic | 369,089 | 252,605 | 181,861 |
| 1.81 | PAFAH1B1 | Platelet-activating factor acetylhydrolase IB subunit alpha | 300,901 | 265,553 | 166,415 |
| 1.69 | TAGLN2 | Transgelin-2 | 1,107,559 | 969,888 | 657,056 |
| 1.38 | RPS12 | 40S ribosomal protein S12 | 737,688 | 710,500 | 534,597 |
| 1.38 | SMYD1 | Histone-lysine N-methyltransferase SMYD1 | 5,684,571 | 4,923,397 | 4,120,347 |

**Fig 3. Overview of layer-specific proteomic profiling of human heart samples. (A)** Principal component analysis based on 1,250 proteins for the inner longitudinal layer (In), 1,217 proteins for the middle circular layer (Mid), and 1,255 proteins for the outer longitudinal layer (Out). Numbers represent the individual cases. The proteomic profiles of SCD and CH cases were distinguishable from those of the control cases in all layers. **(B)** Major proteins with the greatest fold changes between SCD and control cases. Proteins identified with at least two unique peptides with $p$ values $< 0.05$ as determined by the Steel–Dwass test were selected. The top 10 proteins with decreased abundance in each layer of SCD cases are listed. In contrast, only 9, 3, and 7 proteins met the criteria for increased abundance in SCD cases in the respective layers. The fold change indicates the ratio of the relative abundance of SCD to Ctrl for proteins with increased levels, and that of Ctrl to SCD for proteins with decreased levels. Abbreviations: CH, cardiac hypertrophy; Ctrl, control; SCD, sudden cardiac death; PC, principal component.

The decreased proteins in SCD cases compared with control cases, especially in the inner layer, were strongly related to various types of cardiomyopathies, including diabetic, dilated, and arrhythmogenic right ventricular cardiomyopathies (Fig 4B, 4C, S8 Table). In contrast, no gene concept was identified for proteins with increased abundance in either the inner or middle layers in SCD cases, and only seven concepts were identified for increased proteins in the outer layer. However, histological findings of autopsies did not suggest the presence of these cardiomyopathies (Fig 2A). Moreover, critical mutations were not detected in the *PKP2*-coding region, whereas PKP2 levels were transmurally decreased in SCD cases. No GO terms or KEGG pathways were significantly associated with the proteins differentially detected between SCD and CH cases across all layers.

### Decreased Ca$^{2+}$-handling and energy-producing proteins from CH to SCD cases

Levels of cardiomyopathy-related proteins, cadherin-binding proteins, including PKP2, and Ca$^{2+}$-handling proteins such as RYR2, SERCA2a, and NCX gradually decreased from control through CH to SCD groups (Fig 5). In the inner and middle layers, the levels of energy-producing proteins involved in glycolysis and mitochondrial oxidative phosphorylation were lower in the SCD group than in the CH group.

In control hearts, PKP2 was tightly localized at IDs between cardiomyocytes (Fig 6). However, ID detachment was observed in SCD and CH hearts, and the density of PKP2 levels was generally lower in SCD hearts. RYR2 was widely expressed in the myocardial cytoplasm and was highly expressed in IDs of control hearts. In contrast, reduction in RYR2 levels at IDs and in the cytoplasm was observed in SCD and CH hearts. In particular, SCD hearts exhibited a striated decolorization pattern. These microscopic changes were most prominent in the inner myocardial layer.

Abbreviations: CH, cardiac hypertrophy; ID, intercalated disk; SCD, sudden cardiac death

## Discussion

In this study, we showed the spatial proteomic changes in human SCD hearts. We focused on the three myocardial layers—inner longitudinal, middle circular, and outer longitudinal. Laser microdissection enabled layer-specific proteomic profiling of archived human hearts which demonstrated that levels of cardiomyopathy-related and energy-producing proteins were decreased, most significantly in the inner layer of SCD hearts (Fig. 7).

### Downregulation of PKP2 and RYR2

The level of PKP2, a desmosomal protein that maintains intercellular mechanical connections and electrical coupling [20], was transmurally decreased, most prominently in the inner layer of SCD hearts. These immunohistochemical results imply that intercellular adhesion is weakened in SCD hearts.

PKP2 is essential for the normal development of the myocardial junction, as mice with germline PKP2 knockout (KO) present with cytoskeletal disarray and rupture of the cardiac wall, leading to embryonic death [21]. Clinically, mutations in PKP2 are frequently observed in patients with congenital arrhythmias, such as arrhythmogenic cardiomyopathy, Brugada syndrome, and idiopathic ventricular fibrillation [22]. However, none of the enrolled individuals had a family history of

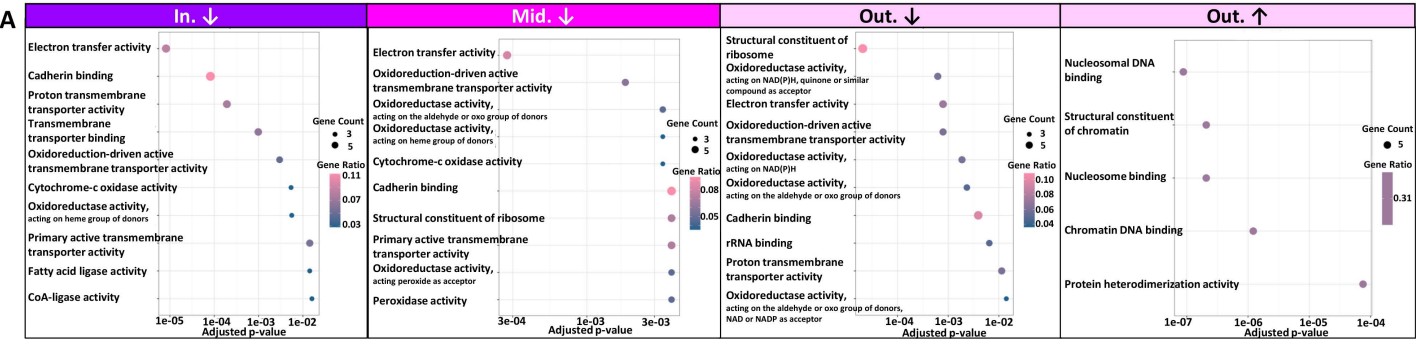

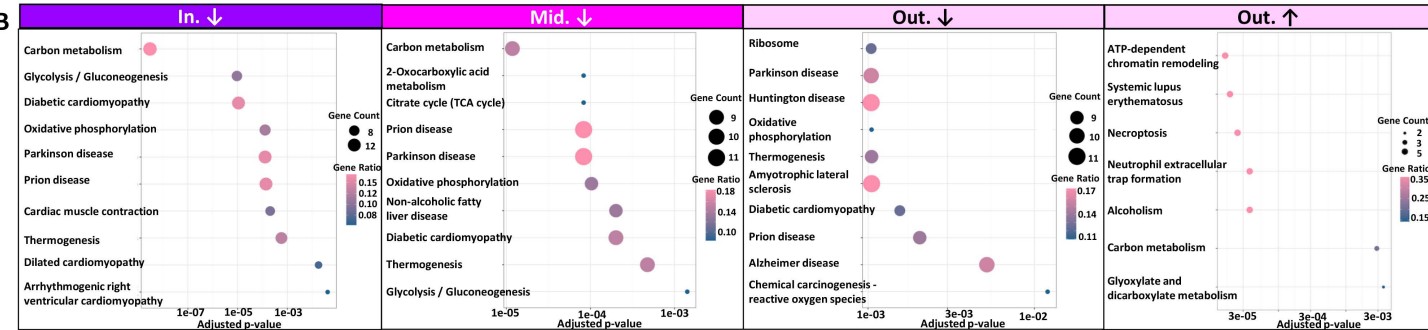

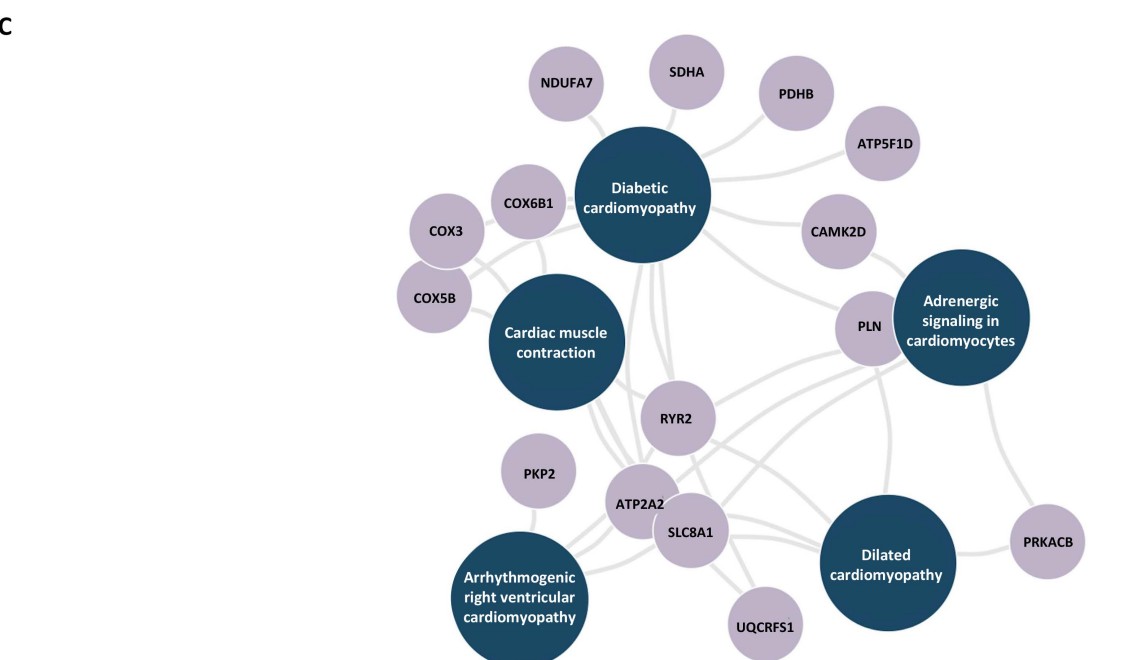

**Fig 4. Bioinformatic analysis of cardiomyocyte proteins with altered abundance in SCD cases compared to control cases. (A)** Dot plots summarizing major GO terms and related molecular functions. Only GO terms that remained significantly overrepresented after adjusting for multiple tests are shown. No GO terms were associated with proteins of increased abundance in either the inner or middle layers in SCD cases. The circle size indicates the number of proteins linked to each GO term. The circle color indicates gene ratio, which indicates the proportion of input genes associated with a specific GO term (number of genes in the term/ total number of input genes). **(B)** Corresponding KEGG pathway analysis for each layer. No significantly enriched pathways were identified for proteins with increased abundance in either the inner or middle layers. **(C)** Gene concept network focusing on cardiovascular diseases, based on proteins with decreased abundance in the inner layer of SCD cases. Abbreviations: GO, gene ontology; SCD, sudden cardiac death.

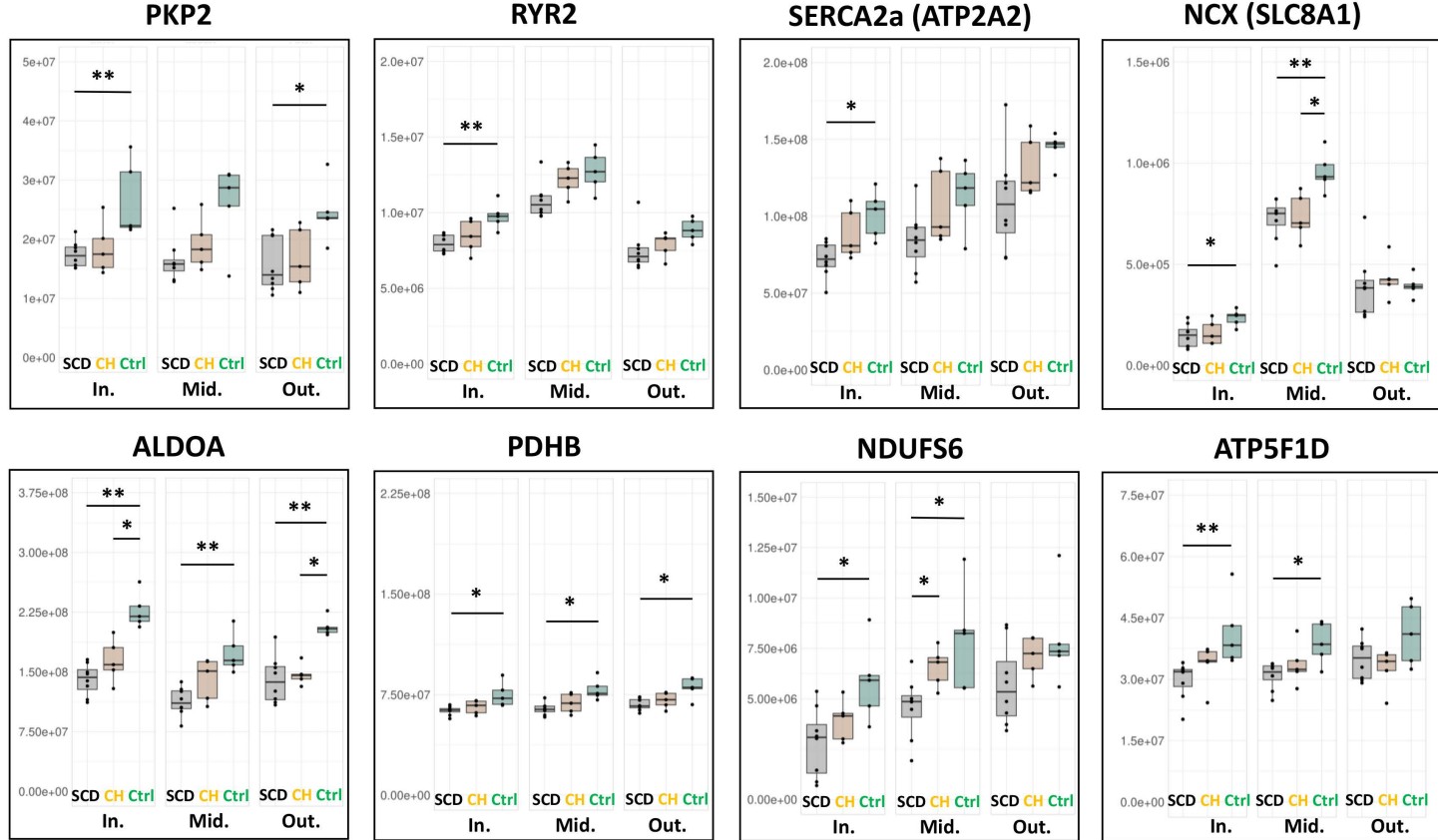

**Fig 5. Proteomic changes in cardiomyopathy-related and energy-producing proteins.** Stepwise alterations in cardiomyocyte protein levels are evident from the control through CH to SCD groups, especially in the inner layer. *$p < 0.05$, **$p < 0.01$ as determined by the Steel-Dwass test. Abbreviations: CH, cardiac hypertrophy; ID, intercalated disk; SCD, sudden cardiac death.

arrhythmia, and critical mutations in *PKP2*-coding regions were not detected. Therefore, decreased PKP2 levels in SCD hearts were most likely induced by acquired pathology, including epigenetic alterations, rather than congenital changes.

Moreover, the downregulation of RYR2, a sarcoplasmic reticulum (SR) transmembrane protein, was observed mainly in the inner layer of SCD hearts. As RYR2 is dense and close to the T-tubule at the Z band, RYR2 attenuation with a striated pattern in SCD hearts may reflect the disturbance of T-tubule structures at the sarcomere. RYR2 releases large amounts of $Ca^{2+}$ from the SR in response to mildly elevated intracellular $Ca^{2+}$ levels induced by L-type $Ca^{2+}$ channel ($Ca_v1.2$) activation during an action potential. The substantial increase in $Ca^{2+}$ concentration ($Ca^{2+}$-induced $Ca^{2+}$ release) caused by RYR2 leads to cardiomyocyte contraction, and thus RYR2 plays a central role in the excitation-contraction coupling of cardiomyocytes. Interestingly, the conditional KO (cKO) of PKP2 in mice reduces the level of RYR2 and enhances its $Ca^{2+}$ sensitivity [23]. Defective RYR2 gating and the associated aberrant $Ca^{2+}$ release from the SR during diastole trigger arrhythmia initiation in PKP2 cKO mice [24,25].

The constitutive KO of RYR2 in mice results in embryonic death with SR vacuolation [26], and RYR2 cKO mice show tachycardic arrhythmia, resulting in SCD [27]. However, cKO rabbits with 50% loss of the RYR2 protein do not develop an overt phenotype in the whole heart with normal levels of other $Ca^{2+}$-handling proteins, although remodeling of RYR2 clusters and significant desynchronization of $Ca^{2+}$ transients have been observed at the myocyte level [28]. These animal experiments demonstrate species specificity in $Ca^{2+}$ homeostasis mechanisms, and the importance of experimental

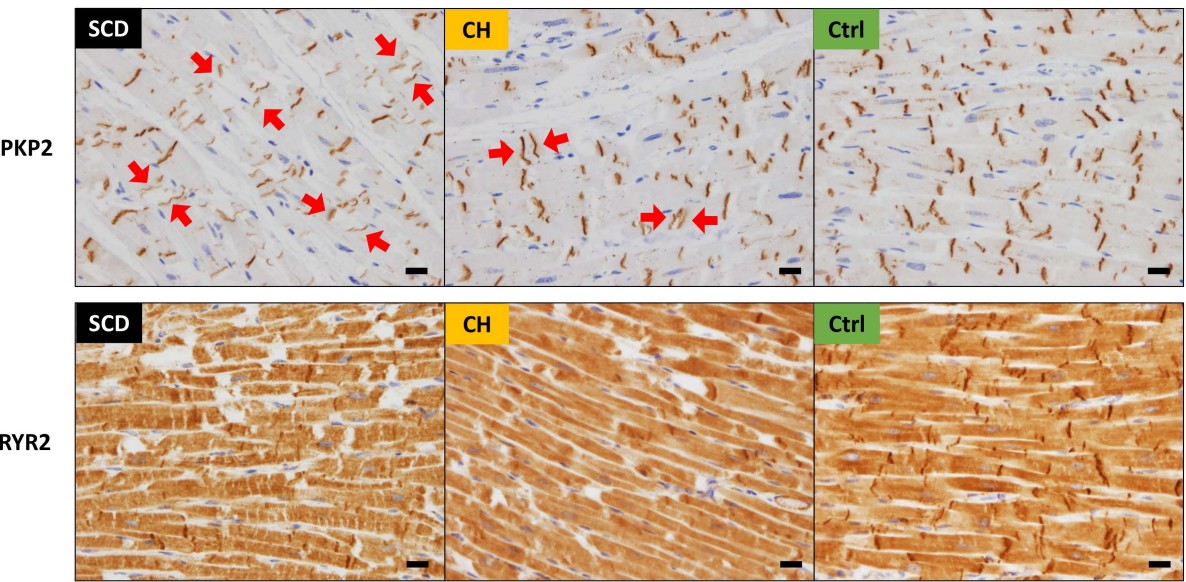

**Fig 6. Representative immunohistochemistry images of the inner myocardial layer.** PKP2, which is localized at the intercalated discs between cardiomyocytes, indicates ID detachment in the SCD and CH hearts (Arrows). RYR2, which is widely expressed in the myocardial cytoplasm and IDs of control hearts, shows a striated decolorization pattern in SCD hearts. To visualize the striated pattern including IDs and T-tubules in the longitudinal sections of cardiomyocytes, the vertical section of the ventricular wall was used for evaluation of the inner longitudinal layer. Scale bar: 20 μm.

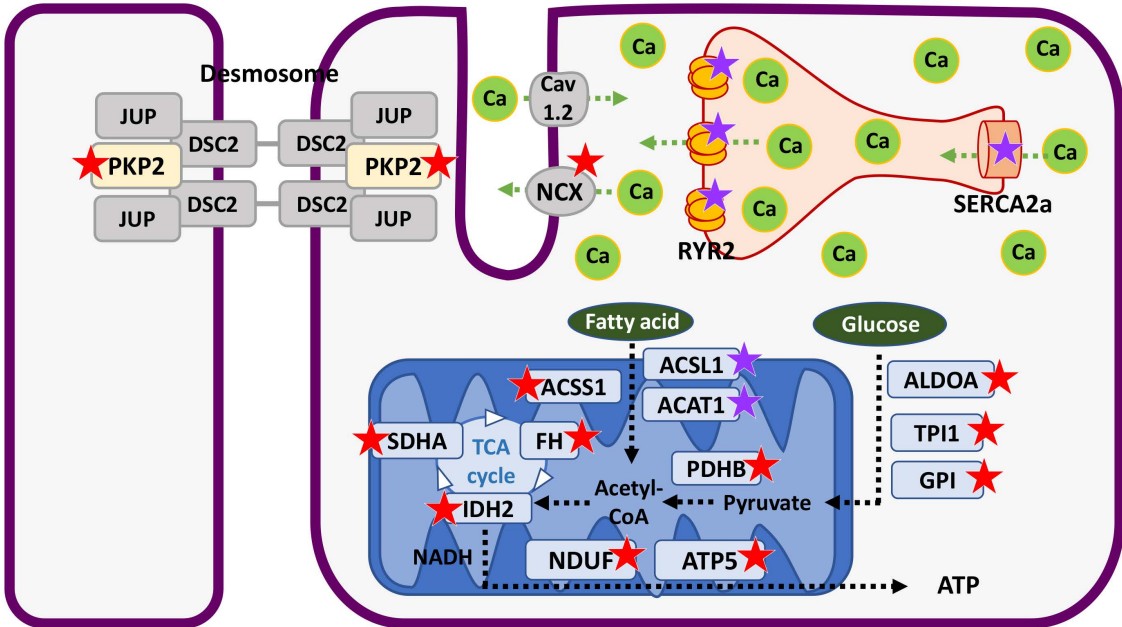

**Fig 7. Summary of the proteomic changes mainly in the inner layer of SCD hearts.** Many proteins that contribute to calcium handling and energy production are decreased. Red stars indicate proteins with significantly decreased levels in the inner and other layers of SCD hearts. Purple stars indicate proteins with significantly decreased levels only in the inner layer of SCD hearts. Abbreviation: SCD, sudden cardiac death.

validation in the human heart to evaluate the pathogenicity of altered protein levels. Moreover, different results at the levels of the entire heart and the cardiomyocytes indicate the necessity of comprehensive expression profiling of $Ca^{2+}$-handling proteins using human tissue samples to reveal the changes underlying pathological cardiac excitation-contraction coupling.

## Downregulation of SERCA2a and NCX

Proteomic profiling of human SCD hearts also revealed the downregulation of $Ca^{2+}$-handling proteins, such as SERCA2a and NCX. SERCA2a is a transmembrane protein that transports $Ca^{2+}$ back into the SR, whereas NCX is a cell membrane-bound protein that excretes $Ca^{2+}$ from the cell. These proteins have a diastolic effect on the human heart; approximately 76% and 24% of $Ca^{2+}$ are removed by SERCA2a and NCX, respectively [29]. Cardiac SERCA2a abundance correlates with PKP2 abundance [30], and a mild decrease in SERCA2a has been observed in PKP2-mutant cardiomyocytes [31]. The downregulation of SERCA2a induces ventricular arrhythmia during ischemia-reperfusion [32] and oxidative stress [33]. Therefore, the decreased SERCA2a levels observed in this study may be related to PKP2 downregulation and an increased risk of ventricular arrhythmia. Moreover, decreased NCX activity has been reported to promote proarrhythmic delayed afterdepolarization and RYR2 gating failure in PKP2 cKO mice [25]. As NCX is concentrated in the T-tubular membrane, its blockade increases $Ca^{2+}$ levels in the dyadic cleft (T-tubule–SR junction) and triggers $Ca^{2+}$ sparks [34]. Therefore, decreased NCX levels in SCD hearts may also contribute to intracellular $Ca^{2+}$ overload and increase the risk of ventricular arrhythmia.

## Downregulation of energy-producing proteins

In this study, the levels of various energy-producing proteins were transmurally decreased in SCD hearts. A reduction of energy-producing proteins including glycolysis-related proteins, such as ALDOA and TPI1 [35], and mitochondrial proteins, such as oxidative phosphorylation complexes [36], has also been reported in the arterial tissue of patients with atrial fibrillation. These proteins were most substantially decreased in the inner layer of SCD hearts.

Mitochondrial dysfunction with the downregulation of electron transport activity facilitates the production of reactive oxygen species (ROS) and induces ventricular arrhythmias during ischemia-reperfusion injury [37]. Mitochondrial ROS production promotes $Ca^{2+}$ leakage from the SR through RYR2 [38] and triggers a transient increase in $Ca^{2+}$ sparks [39]. In contrast, increased oxidative stress inhibits SERCA2a-dependent $Ca^{2+}$ transport into the SR and enhances intracellular $Ca^{2+}$ overload [40]. Moreover, ROS promote reverse-mode NCX transport, whereby $Na^+$ efflux and $Ca^{2+}$ influx in ischemia-reperfusion contribute to $Ca^{2+}$ overload [41]. Thus, mitochondrial dysfunction increases the risk of arrhythmia through enhanced ROS generation and reduced ATP production, necessary for the maintenance of ion channels and transporters [42].

## Impact of endocardial ischemia

Based on physiological hemodynamics of the heart, chronic ischemia promotes myocardial fibrosis from the inner to the outer layer, with the progression of coronary atherosclerosis or cardiac hypertrophy. A pathological study focusing on the three myocardial layers in relation to cardiac strain reported that subendocardial fibrosis was weakly correlated with a decrease in the longitudinal strain of the hypertensive rat heart [43]. In contrast, our study revealed an inner-layer-dominant proteomic alteration without significant myocardial fibrosis, indicating that proteomic changes precede myocardial fibrosis in chronic mild ischemia.

Regarding the effects of ischemia on the expression of individual proteins, it has been reported that cardiac PKP2 levels are decreased in models of ischemia-reperfusion injury [44] and myocardial infarction [45]. SERCA2a suppression is a hallmark of ischemic heart disease [46,47] and cardiac hypertrophy [48]. Moreover, mitochondrial oxidative phosphorylation of proteins is reduced in rats with ischemic heart failure [49] and in a model of hypertensive heart failure [50]. Thus, the proteomic changes observed mainly in the inner layer were at least partly caused by endocardial ischemia.

Overall, the detected levels of each protein were moderately changed in SCD hearts, generally less than two-fold compared with the levels in control hearts. However, the combination of moderate changes in several Ca²⁺-handling proteins with mitochondrial dysfunction increases the risk of arrhythmia, and inner longitudinal layer dominance contributes to the decreased longitudinal strain often observed in high-risk patients with SCD.

## Implications for forensic and clinical practice

Currently, cardiac injury markers used in clinical settings, such as cardiac troponin T and CK-MB in blood, are known to increase nonspecifically after death, making them unreliable for postmortem diagnosis. Furthermore, serum samples are often unavailable in postmortem investigations, and histological examination remains the cornerstone for determining the cause of death.

In this study, histological analysis revealed no significant myocardial fibrosis, and proteomic analysis did not detect significant alterations in known cardiac injury markers in SCD hearts. This may be explained by the fact that the SCD cases in this study did not suffer chronic heart failure and experienced fatal arrhythmia leading to cardiac arrest within an extremely short period, without histological evidence of myocardial necrosis. In contrast, the proteomic alterations identified in the inner myocardial layer in this study, which may lead to arrhythmic death, could precede myocardial fibrosis. These alterations can be detected by routine laboratory methods such as immunohistochemistry and may serve as histological indicators suggesting arrhythmogenic susceptibility. Although SCD remains a diagnosis of exclusion after ruling out other causes, detection of these protein changes may improve the diagnostic accuracy of SCD.

From a clinical perspective, measurement of the identified protein markers in the blood may offer a novel approach for early risk stratification of SCD before onset. Although circulating protein levels were not examined in the present study, the proteomic alterations identified in the inner myocardial layer could influence their circulating concentrations. Notably, serum BNP levels—known to correlate with SCD risk in patients with chronic heart failure—were not elevated in the SCD group of this study. These findings suggest that the identified protein candidates may serve as even earlier, subclinical indicators of susceptibility to SCD, preceding conventional biomarkers such as BNP.

## Study limitations

During sample collection, we selected fresh tissue specimens at autopsy, within 24 hours of death. However, some degree of postmortem degradation is unavoidable when using human autopsy tissues, and this may have affected proteomic profiling. In addition, microscopical tissue sampling required considerable time, resulting in a relatively small sample size in this study. Moreover, clinical history—including comorbidities, daily medication use, and data at the time of death—was not always obtained after death. Therefore, while the findings are clinically valuable, they should be regarded as observational and hypothesis-generating rather than definitive.

In this study, high-throughput MS was used to quantify thousands of proteins; however, the protein levels do not directly reflect protein activity *in vivo.* For example, decreased RYR2 levels result in increased Ca²⁺ release through RYR2 [23]. Therefore, the decreased protein levels detected in the current study do not necessarily indicate decreased activity of this protein. As we used FFPE-archived tissues, we could not directly evaluate the activity of each protein in this study. Fresh samples enable the evaluation of protein activity; however, layer-specific cardiomyocyte sampling without fixation is difficult. Thus, protein activity assays for myocardial layers are a future topic of SCD research.

Moreover, protein modifications, including phosphorylation, glycosylation, ubiquitination, and oxidation, which are important regulators of protein activity, have not been examined. For example, phosphorylation and oxidation increase the open probability of RYR2 and promote Ca²⁺ leakage from RYR2, resulting in enhanced arrhythmic susceptibility [51]. Future comprehensive studies on protein modifications will deepen our understanding of post-translational alterations in SCD in middle-aged and older individuals.

## Conclusion

This is the first report on the layer-specific proteomic profiling of human SCD hearts. Arrhythmogenic changes, including reduced levels of PKP2, RYR2, SERCA2a, NCX, and mitochondrial proteins, develop most prominently in the inner layer. As the proteome profile gradually changes from CH to SCD before myocardial fibrosis progresses, the detection of endocardial proteomic alterations may aid in the postmortem diagnosis of SCD in middle-aged and older asymptomatic individuals with CH.

### Highlights

- Human cardiomyocytes were microdissected separately from the three myocardial layers.
- Proteomic alterations were most pronounced in the inner layer in sudden cardiac death (SCD).
- SCD hearts exhibited decreased levels of energy-producing and $Ca^{2+}$-handling proteins.
- Proteomic changes progressed stepwise from control to cardiac hypertrophy and SCD.
- These proteomic changes appeared before the development of myocardial fibrosis.

## Supporting information

**S1 Table. Oligonucleotide sequences for PKP2.**
(DOCX)

**S2 Table. Inter-group comparisons in all cases.**
(XLSX)

**S3 Table. Inter-layer comparisons in SCD cases.**
(XLSX)

**S4 Table. Inter-layer comparisons in CH cases.**
(XLSX)

**S5 Table. Inter-layer comparisons in control cases.** Note that the original data for inter-layer comparisons in control cases was previously published [13]; in the present study, we reanalyzed these data using multiple nonparametric methods.
(XLSX)

**S6 Table. Differentially expressed protein lists.**
(XLSX)

**S7 Table. Gene Ontology (GO) analysis of proteins with altered levels in SCD cases.**
(XLSX)

**S8 Table. Pathway analysis of proteins with altered levels in SCD cases.**
(XLSX)

**S1 File. SuppData** .
(DOCX)

## Acknowledgments

The authors are grateful to the technical staff at the Forensic Laboratory and Support Center for Medical Research and Education at Tokai University.

## Author contributions

**Conceptualization:** Yu Kakimoto.

**Data curation:** Yu Kakimoto, Xueting Guan.

**Funding acquisition:** Yu Kakimoto.

**Investigation:** Yu Kakimoto, Xueting Guan, Atsushi Ueda, Yayoi Kimura, Tomoko Akiyama, Masayuki Tanaka.

**Resources:** Haruka Ikeda.

**Supervision:** Kazuho Maeda, Shotaro Isozaki.

**Writing – original draft:** Yu Kakimoto, Xueting Guan, Atsushi Ueda, Yayoi Kimura, Tomoko Akiyama.

**Writing – review & editing:** Masayuki Tanaka, Haruka Ikeda.

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
