## [Decision Letter · Decision Letter 0]

22 Oct 2025

Dear Dr. Kakimoto,

We look forward to receiving your revised manuscript.

Kind regards,

Aldrin V. Gomes, Ph.D.

Academic Editor

PLOS ONE

Journal Requirements:

“This work was supported by JSPS KAKEHNHI (grant number: 23H03178) and the MEXT promotion of the Distinctive Joint Research Center Program at the Advanced Medical Research Center, Yokohama City University (grant number: JPMXP0622717006).”

4. Please note that funding information should not appear in any section or other areas of your manuscript. We will only publish funding information present in the Funding Statement section of the online submission form. Please remove any funding-related text from the manuscript.

5. Thank you for uploading your study's underlying data set. Unfortunately, the repository you have noted in your Data Availability statement does not qualify as an acceptable data repository according to PLOS's standards.

Additional Editor Comments: 

The reviewers all commented positively on your manuscript. However, several concerns need to be addressed including the need for within-group comparisons of proteomic profiles across three ventricular layers, particularly in the SCD group. 

Reviewer's Responses to Questions

**Comments to the Author**

1. Is the manuscript technically sound, and do the data support the conclusions?

Reviewer #1: Yes

Reviewer #2: Partly

Reviewer #3: Yes

2. Has the statistical analysis been performed appropriately and rigorously?

Reviewer #1: Yes

Reviewer #2: No

Reviewer #3: Yes

3. Have the authors made all data underlying the findings in their manuscript fully available?

Reviewer #1: Yes

Reviewer #2: Yes

Reviewer #3: Yes

4. Is the manuscript presented in an intelligible fashion and written in standard English?

Reviewer #1: Yes

Reviewer #2: Yes

Reviewer #3: Yes

Reviewer #1: The paper titled: “Layer-specific proteomic analysis of human hearts in patients with sudden cardiac death” is authored by Yu Kakimoto et al.

Some concerns could be addressed to consolidate the scientific message:

1- Please indicate the scale bar on each picture of Figure 2 and 5.

2- Indicate the dots in the whisker box plot graphs.

3- Figure legends must be consolidated with a scientific title and more descriptive information. Please improve.

4- Bar graph from figure 5 are unreadable. Please separate in different figures and increase the size of graphs.

5- Define all abbreviations at first use.

6- More information about the included patients is required, including sex, comorbidities, medication, weight, etc.

7- Provide a Highlight section with up to 5 bullet points to describe the main findings of the paper.

Reviewer #2: Review Comments to the Author

This study, based on sudden cardiac death (SCD) cases and matched controls, presents meaningful findings. However, the overall presentation and interpretation of the data are insufficient in several key aspects.

Merits:

The research team enrolled SCD cases and controls over a nearly 10-year period and compared proteomic features across the three ventricular layers-an approach that adds spatial resolution and biological insight to the analysis.

Shortcomings and Recommendations:

1. Layer-Specific Comparison Within Groups

I strongly recommend that the authors perform within-group comparisons of proteomic profiles across the three ventricular layers, particularly in the SCD group. This would help clarify whether layer-specific proteomic alterations are associated with the pathophysiology of SCD.

2. Validation of Layer-Specific Protein Expression

It would be informative to simultaneously demonstrate the expression of validated proteins (e.g., PKP2, RyR2) across all three layers within individual samples. This would allow for a direct assessment of inter-layer expression heterogeneity and strengthen the spatial proteomic

3. Protein Degradation analysis

Postmortem autolysis can lead to significant protein degradation, potentially confounding proteomic findings. The authors should carefully evaluate and, if possible, quantify the extent of postmortem protein degradation to ensure that observed differences are primarily disease-related rather than degradation-related.

4. Data Presentation and Statistical Clarity

The result section: each subtitle should represent the main finding of the relative part. For example, line 238, this title meaningless and need to be modified according to the relative contents or findings.

Figure 1: The BNP levels appear markedly different between SCD and control groups. Please clarify why this difference was not statistically significant.

Figure 2: Provide detailed methodology for myocyte diameter measurement. How was measurement consistency ensured across different regions? Also, report the total number of myocytes analyzed per group.

Figure 3B: The fold changes do not appear to correspond directly to the relative abundance values. Please verify whether fold changes should be expressed as inverse values (e.g., 1/9.2 ≈ 0.11). Additionally, comparisons between SCD and CH (control heart?) groups should be included.

Figure 4A-C: Similar to Figure 3B, include direct comparisons between SCD and CH groups to enhance interpretability.

Figure 5B: The immunostaining images lack clarity in distinguishing target signals (e.g., PKP2, RyR2) from background. Consider adjusting color contrast or providing higher-resolution images. Please also specify the sample size for each group and include a comprehensive statistical summary table.

SuppData: I suggested to give a full list of differential expressed proteins (DEPs) among different groups and among different layers. Each data table should have a specific name.

5. Discussion-Forensic Application Potential

The discussion should be expanded to address the potential utility of the identified proteins in forensic practice. A comparative analysis with existing cardiac biomarkers used in postmortem SCD diagnosis would strengthen the translational relevance of the findings.

Reviewer #3: This Reviewer rewards the authors for this elegant study. It is well written, clear and concise, with novel and interesting data on the proteomic changes in the three muscular layers of the human heart in cases of sudden cardiac death and cardiac hypertrophy (w/o HCM).

**Do you want your identity to be public for this peer review?** For information about this choice, including consent withdrawal, please see our Privacy Policy

Reviewer #1: No

Reviewer #2: No

Reviewer #3: **Yes: ** Ida Gjervold Lunde

---

## [Author Response · Author response to Decision Letter 1]

3 Nov 2025

Journal Requirements:

Response: We have checked the PLOS ONE style templates.

Response: We have ensured that JSPS KAKEHNHI (grant number 23H03178) and the MEXT promotion of the Distinctive Joint Research Center Program at the Advanced Medical Research Center, Yokohama City University (grant number JPMXP0622717006) are correct.

“This work was supported by JSPS KAKEHNHI (grant number: 23H03178) and the MEXT promotion of the Distinctive Joint Research Center Program at the Advanced Medical Research Center, Yokohama City University (grant number: JPMXP0622717006).”

Response: We have added the above statement to the cover letter.

4. Please note that funding information should not appear in any section or other areas of your manuscript. We will only publish funding information present in the Funding Statement section of the online submission form. Please remove any funding-related text from the manuscript.

Response: We have removed the funding statement section from the manuscript.

5. Thank you for uploading your study's underlying data set. Unfortunately, the repository you have noted in your Data Availability statement does not qualify as an acceptable data repository according to PLOS's standards.

Response: Thank you for your request. We have deposited the raw data set to Dryad. The stable access information is:DOI: 10.5061/dryad.q573n5tx2.

Reviewer Sharing Link (for private access): http://datadryad.org/share/LINK_NOT_FOR_PUBLICATION/oZ7H4y_TtOndYYe7L2co_qQCkN6W12dxfwaDnjxDv6A.

Response: We have added the captions for supplemental tables at the end of the manuscript.

Response: OK.

Additional Editor Comments:

The reviewers all commented positively on your manuscript. However, several concerns need to be addressed including the need for within-group comparisons of proteomic profiles across three ventricular layers, particularly in the SCD group.

Reviewer's Responses to Questions

Comments to the Author

1. Is the manuscript technically sound, and do the data support the conclusions?

Reviewer #1: Yes

Reviewer #2: Partly

Reviewer #3: Yes

2. Has the statistical analysis been performed appropriately and rigorously?

Reviewer #1: Yes

Reviewer #2: No

Reviewer #3: Yes

3. Have the authors made all data underlying the findings in their manuscript fully available?

Reviewer #1: Yes

Reviewer #2: Yes

Reviewer #3: Yes

4. Is the manuscript presented in an intelligible fashion and written in standard English?

Reviewer #1: Yes

Reviewer #2: Yes

Reviewer #3: Yes

5. Review Comments to the Author

Reviewer #1: The paper titled: “Layer-specific proteomic analysis of human hearts in patients with sudden cardiac death” is authored by Yu Kakimoto et al.

Some concerns could be addressed to consolidate the scientific message:

1- Please indicate the scale bar on each picture of Figure 2 and 5.

Response: Thank you for your scientific review comments. We have revised the manuscript and figures according to your advice. We have added the scale bars.

2- Indicate the dots in the whisker box plot graphs.

Response: Thank you for your helpful suggestion. We have added the data points (dots) to the whisker box plots in the revised Figures 2 and 5.

3- Figure legends must be consolidated with a scientific title and more descriptive information. Please improve.

Response: We have enriched the legends of Fig.2, 3, 5, 6 and 7.

4- Bar graph from figure 5 are unreadable. Please separate in different figures and increase the size of graphs.

Response: Thank you for your feedback regarding the readability of Figure 5. We have separated the original panels 5A and 5B into new Figures 5 and 6. In addition, we have increased the overall graph and font sizes, particularly for the y-axis labels, in the new Figure 5 to improve clarity. The figure numbers have been revised throughout the manuscript accordingly.

5- Define all abbreviations at first use.

Response: We have confirmed that all abbreviations are defined upon their first use.

6- More information about the included patients is required, including sex, comorbidities, medication, weight, etc.

Response: Thank you for your careful review. At the time of forensic autopsy, the clinical history of the deceased is often unknown, and unfortunately, information on medication was unavailable for this study. We have, however, incorporated the available information, including sex, body weight, and comorbidities, into Fig. 1.

7- Provide a Highlight section with up to 5 bullet points to describe the main findings of the paper.

Response: We have added a Highlight section on page 2.

Reviewer #2: Review Comments to the Author

This study, based on sudden cardiac death (SCD) cases and matched controls, presents meaningful findings. However, the overall presentation and interpretation of the data are insufficient in several key aspects.

Merits:

The research team enrolled SCD cases and controls over a nearly 10-year period and compared proteomic features across the three ventricular layers-an approach that adds spatial resolution and biological insight to the analysis.

Response: Thank you for your scientific review comments. We have revised the manuscript and figures according to your advice.

Shortcomings and Recommendations:

1. Layer-Specific Comparison Within Groups

I strongly recommend that the authors perform within-group comparisons of proteomic profiles across the three ventricular layers, particularly in the SCD group. This would help clarify whether layer-specific proteomic alterations are associated with the pathophysiology of SCD.

Response: Thank you for your suggestion. As you recommended, we have performed additional within-group comparisons of proteomic profiles across the three ventricular layers. These results are presented in Supplementary Tables S3 (SCD cases), S4 (CH cases), and S5 (Control cases). This analysis was conducted using the Friedman test, followed by the Wilcoxon signed-rank test for post-hoc comparisons.

2. Validation of Layer-Specific Protein Expression

It would be informative to simultaneously demonstrate the expression of validated proteins (e.g., PKP2, RyR2) across all three layers within individual samples. This would allow for a direct assessment of inter-layer expression heterogeneity and strengthen the spatial proteomic

Response: We agree that demonstrating protein expression across all three layers within individual samples is informative. However, presenting the complete dataset of individual protein profiles in the main text would be excessively lengthy and redundant given the aggregated data already provided. To ensure clarity and maintain focus, we have included these detailed data in Supplementary Table S2.

3. Protein Degradation analysis

Postmortem autolysis can lead to significant protein degradation, potentially confounding proteomic findings. The authors should carefully evaluate and, if possible, quantify the extent of postmortem protein degradation to ensure that observed differences are primarily disease-related rather than degradation-related.

Response: We acknowledge that postmortem degradation is an inevitable limitation when using autoptic tissue samples, and we have addressed this in the limitation section. Unfortunately, the small sample volume of the FFPE tissues precludes us from performing additional validation experiments, such as Western blotting.

4. Data Presentation and Statistical Clarity

The result section: each subtitle should represent the main finding of the relative part. For example, line 238, this title meaningless and need to be modified according to the relative contents or findings.

Response: Thank you for your helpful comments. We have revised all subtitles in the Results section to clearly reflect the main findings of the respective content, and the paragraphs have been rearranged accordingly.

Figure 1: The BNP levels appear markedly different between SCD and control groups. Please clarify why this difference was not statistically significant.

Response: We appreciate your important comments. We apologize for the mispresentation: several values in Fig. 1 incorrectly displayed as standard error (SE). We have corrected this, and Fig. 1 now consistently presents all values as mean ± SD. Due to the notably wide standard deviation (for example, 13.6 ±18.0), a statistically significant difference in BNP levels was not observed between the groups.

Figure 2: Provide detailed methodology for myocyte diameter measurement. How was measurement consistency ensured across different regions? Also, report the total number of myocytes analyzed per group.

Response: As the nucleus is located at the center of each cardiomyocyte, the cardiomyocyte diameter was defined as the minor axis measured at the nuclear level. Because some degree of inconsistency is inevitable in microscopic measurements, we assessed 10 cardiomyocytes per field and averaged the values from 10 randomly selected fields for each case. In total, 100 cardiomyocytes were analyzed per myocardial layer per case. The Methods section has been revised accordingly to include these details.

Figure 3B: The fold changes do not appear to correspond directly to the relative abundance values. Please verify whether fold changes should be expressed as inverse values (e.g., 1/9.2 ≈ 0.11).

Response: The fold change values are calculated to always be greater than 1. Specifically, the fold change represents the ratio of SCD/Ctrl for proteins with increased abundance, and the ratio of Ctrl/SCD for proteins with decreased abundance. We have updated the figure legend to clarify this calculation method.

Additionally, comparisons between SCD and CH (control heart?) groups should be included.

Response: As shown in Table S2, the differences between SCD and CH cases were minimal. Only 17 (10 increased and 7 decreased), 12 (9 increased and 3 decreased), and 13 (5 increased and 8 decreased) proteins were differentially detected in the inner, middle, and outer layers, respectively. We have enriched the result section accordingly.

Figure 4A-C: Similar to Figure 3B, include direct comparisons between SCD and CH groups to enhance interpretability.

Response: We confirm that the number of differentially detected proteins between the SCD and CH groups was small. Consequently, no GO terms or KEGG pathways were significantly associated with the proteins differentially detected between SCD and CH cases across all layers. We have added this clarifying statement to the Results section.

Figure 5B: The immunostaining images lack clarity in distinguishing target signals (e.g., PKP2, RyR2) from background. Consider adjusting color contrast or providing higher-resolution images. Please also specify the sample size for each group and include a comprehensive statistical summary table.

Response: We increased the color contrast by 10% for all figures. We performed immunostaining on four SCD cases, three CH cases, and three control cases, from which a vertical section of the ventricular wall was available. Absolute quantification of these immunostaining images is inherently difficult, and moreover, the small sample size makes reliable statistical analysis infeasible. Consequently, we did not perform statistical tests on the immunostaining results. We have revised the Methods section accordingly.

SuppData: I suggested to give a full list of differential expressed proteins (DEPs) among different groups and among different layers. Each data table should have a specific name.

Response: Thank you for the suggestion. Comprehensive lists of differentially expressed proteins (DEPs) have been compiled in S6 Table, each with specific nomenclature. DEPs among groups were identified using the Kruskal–Wallis test (p < 0.05), and DEPs among layers within each group were identified using the Friedman test (p < 0.05). The table includes complete protein annotations and detailed statistical results, including post hoc p-values obtained from the Steel–Dwass and Wilcoxon signed-rank tests, respectively.

5. Discussion-Forensic Application Potential

The discussion should be expanded to address the potential utility of the identified proteins in forensic practice. A comparative analysis with existing cardiac biomarkers used in postmortem SCD diagnosis would strengthen the translational relevance of the findings.

Response: We appreciate the reviewer’s

---

## [Decision Letter · Decision Letter 1]

14 Dec 2025

Layer-specific proteomic analysis of human hearts in patients with sudden cardiac death

PONE-D-25-50384R1

Dear Dr. Kakimoto,

We’re pleased to inform you that your manuscript has been judged scientifically suitable for publication and will be formally accepted for publication once it meets all outstanding technical requirements.

Kind regards,

Aldrin V. Gomes, Ph.D.

Academic Editor

PLOS One

Additional Editor Comments (optional):

Reviewers' comments:

Reviewer's Responses to Questions

**Comments to the Author**

Reviewer #1: All comments have been addressed

Reviewer #2: All comments have been addressed

2. Is the manuscript technically sound, and do the data support the conclusions?

Reviewer #1: Yes

Reviewer #2: Yes

3. Has the statistical analysis been performed appropriately and rigorously?

Reviewer #1: Yes

Reviewer #2: Yes

4. Have the authors made all data underlying the findings in their manuscript fully available?

Reviewer #1: Yes

Reviewer #2: Yes

5. Is the manuscript presented in an intelligible fashion and written in standard English?

Reviewer #1: Yes

Reviewer #2: Yes

Reviewer #1: The authors have addressed my comments correctly.

I do not have additional concerns.

The authors have addressed my comments correctly.

I do not have additional concerns.

Reviewer #2: All my concerns have been addressed, and I have no more conern. I think the manuscript could be published at its current version.

**Do you want your identity to be public for this peer review?** For information about this choice, including consent withdrawal, please see our Privacy Policy

Reviewer #1: No

Reviewer #2: No

---

## [Editor Report · Acceptance letter]

PONE-D-25-50384R1

PLOS One

Dear Dr. Kakimoto,

I'm pleased to inform you that your manuscript has been deemed suitable for publication in PLOS One. Congratulations! Your manuscript is now being handed over to our production team.

Kind regards,

on behalf of

Dr. Aldrin V. Gomes

Academic Editor

PLOS One